# A systematic approach to study the effects of acquisition parameters and biological factors on computerized mammography analysis using ex vivo human tissue: A protocol description

Nicole Hernández [1,2]*, Tomppa Pakarinen[1,3], Annukka Salminen[1,2], Santiago Laguna Castro[4,5], Ulla Karhunen-Enckell[1,6], Markus Hannula[4,5], Ritva Heljasvaara[7], Jari Hyttinen [4,5], Katriina Joensuu[6,8], Otto Jokelainen[9,10], Arja Jukkola[11,12], Sanna-Maria Karppinen[7], Auni Lindgren[13,14], Eero Lääperi[1], Emilia Peuhu[15,16,17], Taina Pihlajaniemi[7], Renata Prunskaite-Hyyryläinen[7], Kirsi Rilla[18], Pekka Ruusuvuori[1,19], Leena Latonen[18,19], Teemu Tolonen[20], Masi Valkonen[19], Mira Valkonen[1,19], Miska Vuorlaakso[1,21], Said Pertuz[22], Irina Rinta-Kiikka[1,2], Otso Arponen [23]

1 Faculty of Medicine and Health Technology, Tampere University, Tampere, Finland, 2 Department of Radiology, Tampere University Hospital, Tampere, Finland, 3 Department of Medical Physics, The Wellbeing Services County of Pirkanmaa, Tampere, Finland, 4 Computational Biophysics and Imaging Group, Faculty of Medicine and Health Technology, Tampere University, Tampere, Finland, 5 BioMediTech Unit, Faculty of Medicine and Health Technology, Tampere University, Tampere, Finland, 6 Department of Surgery, Tampere University Hospital, Tampere, Finland, 7 Faculty of Biochemistry and Molecular Medicine, University of Oulu, Oulu, Finland, 8 Department of Plastic Surgery, Tampere University Hospital, Tampere, Finland, 9 Institute of Clinical Medicine, Pathology and Forensic Medicine, University of Eastern Finland, Kuopio, Finland, 10 Department of Clinical Pathology, Diagnostic Imaging Center, Kuopio University Hospital, Kuopio, Finland, 11 Department of Oncology, Tampere University Hospital, Tampere, Finland, 12 Cancer Center, Faculty of Medicine and Health Technology, Tampere University, Tampere, Finland, 13 Department of Gynecology and Obstetrics, Kuopio University Hospital, Kuopio, Finland, 14 Institute of Clinical Medicine, University of Eastern Finland, Kuopio, Finland, 15 Institute of Biomedicine, Cancer Laboratory FICAN West, University of Turku, Turku, Finland, 16 Turku Bioscience Centre, University of Turku, Turku, Finland, 17 Åbo Akademi University, Turku, Finland, 18 Institute of Biomedicine, University of Eastern Finland, Kuopio, Finland, 19 Institute of Biomedicine, University of Turku, Turku, Finland, 20 Department of Pathology, Fimlab Laboratories, Tampere, Finland, 21 Department of Musculoskeletal Surgery and Diseases, Tampere University Hospital, Tampere, Finland, 22 Department of Electrical, Electronics and Telecommunications Engineering, Universidad Industrial de Santander, Bucaramanga, Colombia, 23 Institute of Clinical Medicine, University of Eastern Finland, Kuopio, North Savo, Finland

☯ These authors contributed equally to this work.
* nicole.hernandezduran@tuni.fi

**Data availability statement:** No datasets were generated or analysed during the current study.

## Abstract

**Background:** Mammography is the most common imaging modality for the detection of breast cancer. Artificial intelligence algorithms for mammography analysis have shown promising performance for breast cancer risk assessment and lesion detection and classification; however, these models often fail the test of external validation. The evidence points to variations in image acquisition—known as the batch effect—as a main

All relevant data from this study will be made available upon study completion.

**Funding:** This work is supported by the following grants, awarded to O.A.: seed funding for Health Data Science projects scheme of Tampere University, Tampere University Hospital (Project No. MJ006L), the Competitive State Research Financing of the Expert Responsibility Area of Tampere University Hospital (Project Nos. 9AC002 and T62564), the Cancer Foundation Finland and the Finnish Medical Foundation. The funders had no role in the design of this work, its execution, analyses, interpretations, or the decision to publish the article.

**Competing interests:** The authors have declared that no competing interests exist.

contributing factor to the lack of the models generalization and robustness. However, studies on the effects of acquisition in the mammogram have been limited due to lack of appropriate datasets. This prospective, exploratory, non-randomized study aims to study how biological and non-biological sources of heterogeneity affect the mammogram and, in turn, the computerized models for mammography analysis.

**Methodology:** This study will collect breast samples from 200 participants that will undergo breast resection as per clinical indications. Each sample will undergo the mammography imaging procedure several times to obtain mammograms with different combinations of imaging parameters. The resulting dataset will be used for the statistical analysis of the impact of imaging parameters in mammographic texture features and the computerized analysis of mammograms. Furthermore, biological information will be collected from the resected breast samples to study their relation to mammographic texture features.

**Discussion:** This study will add to the understanding of the effect of different sources of heterogeneity on mammography, ultimately aiding in the future development of robust computerized analysis models.

## Introduction

### Mammographic imaging and the current use of artificial intelligence

With 2.3 million new cases and 685,000 deaths in 2020, breast cancer is the most diagnosed cancer and accounts for most cancer deaths among females globally [1]. Early detection is of key importance for reducing breast cancer-related mortality [2,3], as these cases have a 5-year survival probability of approximately 96% [4]. Mammography screening programs, in which females aged 50 to 69 are invited for a biennial mammogram, are implemented across the EU [5]. Screening plays a significant role in early detection of the disease [6], as it has been shown that the successful implementation of screening programs can reduce breast cancer mortality by approximately 20% [7]. Mammography screening, however, is less effective on breasts with a higher percentage of fibroglandular tissue (FGT)—i.e., mammographically dense breasts—due to the similarity in the radiodensity between FGT and breast tumor tissue leading to reduced contrast between these tissue types [8,9]. The sensitivity of breast cancer detection decreases with increasing breast density, from 85.7% in the lowest density category to 61.0% in the highest density category [10,11]. Mammographically dense breasts (either heterogeneously dense or extremely dense) are found in about 48% of the screened population [12], and research has shown that females with higher breast density have a greater risk of developing breast cancer [11,13]. Indeed, mammographic breast cancer screening is less effective in females with mammographically dense breasts who simultaneously have the highest risk of developing breast cancer.

Computerized analysis of mammographic images by means of artificial intelligence (AI), specifically classical machine learning and deep learning (DL), is an active area of research with applications to different problems, such as breast cancer risk assessment and breast lesion detection and classification [14]. *Radiomics* is the extraction of quantitative features from medical images, with the aim of correlating them to different health conditions or outcomes [15]. AI-driven radiomics can be divided into classical machine learning approaches, which use hand-crafted radiomic features [15], and DL approaches, which implicitly build data-driven image descriptors using DL architectures [16]. Both approaches have shown potential for breast percent density classification [17,18],

breast cancer risk assessment [19,20], and mass detection [21]. Recently, AI-supported mammography readings have been suggested to reduce reader workload [22] and improve lesion detection [23]. Furthermore, radiomic biomarkers have been suggested to reflect the multiscale complexity of tissues ranging from genomic to histological (i.e., molecular and cellular) levels [24–30].

## Artificial intelligence and the current understanding of the factors affecting its performance

Although AI-based mammography analysis has demonstrated great potential, the lack of access to representative datasets [31–34] and model degradation due to data distribution shift [35–39] are commonly reported challenges that are hindering the translation of AI to clinical practice. The high heterogeneity between mammography datasets due to imaging parameters is a recognized challenge for the development of robust computerized analysis tools. Research to understand the effect of these sources of heterogeneity for the goal of computerized assessment is still lacking [35,40,41].

The effect of non-biological sources of heterogeneity—imaging acquisition and the manufacturer's pre- and post-processing—is referred to as the *batch effect*. The batch effect has been shown to introduce changes in computerized image features, which also affect their robustness and reproducibility [42–44]. This in turn is reflected in the performance of breast cancer risk assessment [37,45,46] and breast cancer classification [35] models. On the other hand, the biological properties of the breast, through the interaction with the x-ray radiation, create the radiomic features; the heterogeneity that arises from them is, technically, desired. However, as radiomic biomarkers are not initially derived from biological samples but rather from radiological images, the effect of the interplay between image acquisition and the underlying biology on the performance of AI-based radiomics is unknown [24]. Research into the role of fibroglandular and adipose tissue—from the genetic and molecular levels to cellular and tissue levels—on radiomic features is currently scarce, but it is needed to explain how radiomic features may predict the future development of cancer.

Although the amount of existing clinical mammography data is large, the retrieval of an appropriate dataset to study the sources of heterogeneity mentioned above is infeasible from existing, accessible repositories. As a source of ionizing radiation, mammographic imaging limits the number of times a person can be imaged. As a result, the acquisition parameters in clinical practice used to capture each specific mammogram are optimized for image quality and for as low a radiation dose as is reasonably achievable: this limits the range of acquisition parameters that are employed on the different breasts, which, in turn, limits the possibilities of analyzing the effect of acquisition parameters on computerized analysis of mammograms derived from *in vivo* breasts. A dataset for the study of the sources of heterogeneity would require reimaging of the breast with different acquisition settings. Moreover, the analysis of biological factors requires the collection of tissue samples. Breast tissue is not normally resected unless there is a suspicion of cancer. Previous studies have used phantoms and retrospectively collected mammograms [42,44,45], which can only partially address the association between acquisition parameters and textural features and do not allow for the systematic evaluation of the effect of biological factors on radiomic features.

## Study objectives

Our hypothesis posits that variations in mammography acquisition introduce a measurable batch effect. Under this hypothesis, it is expected that the batch effect significantly affects the

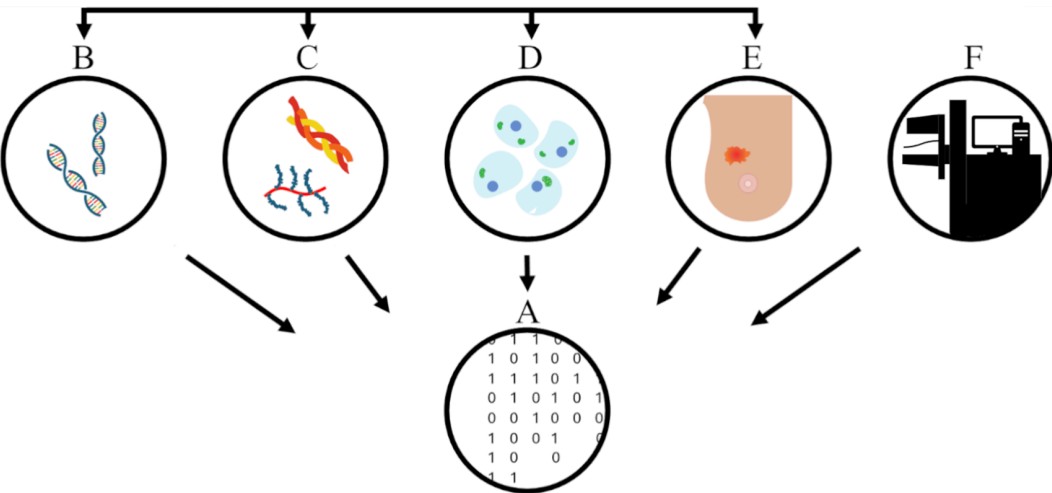

**Fig 1. Factors affecting mammographic texture features.** The study aims to evaluate how mammographic texture features (A) are affected by imaging parameters (F), and by genetic (B), molecular (C), cellular (D), and tissue level (E) biological factors across different scales.

performance and generalizability of AI models for breast cancer detection. The characterization of this effect, as well as a better understanding of the link between biological characteristics and mammographic radiomic features, are fundamental to developing robust models (Fig 1). The study has two objectives:

1. Evaluate how imaging parameters affect the mammographic radiomic features by
   - developing a model for the batch effect in mammography images via quantification of its effect in radiomic features; and
   - assessing the impact of acquisition parameters on the performance of AI models for breast cancer detection from mammography.
2. Evaluate how biological factors across different scales affect computerized mammographic features by
   - assessing associations between different biological characteristics of the breast tissue and radiomic features.

To achieve these goals, we will use *ex vivo* human tissue—resected breasts. This dataset will be characterized by the acquisition of multiple mammograms from the resected breast tissue with different sets of imaging parameters. Such data will allow for the strict study of the interplay between acquisition parameters, breast biology, and mammographic texture features. Furthermore, we will use the acquired dataset to assess the effect of acquisition parameters and biological factors on the performance of breast cancer risk assessment and lesion detection models. In the proposed study, we will overcome the ethical constraints of the repeated use of ionizing radiation with the use of *ex vivo* human tissue for the generation of a dataset suitable for this study. This work aims to contribute to the advancement of computerized mammography analysis and to facilitate its translation into clinical practice through both radiomics and DL-based systems. Equally important, this study aims to provide insight into the associations between clinical and biological factors and

computerized radiomic features. This article primarily details the methodology and analysis pertaining to our first objective, with a focus on the study of the acquisition parameters of mammography.

## Materials and methods

### Methodological design

This is a prospective, exploratory, non-randomized study on adult females (≥18 years old) who will undergo uni- or bilateral mastectomy for clinical indications unrelated to this trial. In this protocol, we broadly define mastectomy as a surgical procedure in which fibroglandular and fatty tissue of the breast is removed. Patients will be recruited at Tampere University Hospital, Finland, the coordinating study center. Multiple study centers in Finland will analyze the resected breast samples.

The multidisciplinary research team (comprising researchers in the fields of surgery, radiology, pathology, oncology, medical physics, computer sciences, and biological sciences) enriches the methodology and scope of this study. The methodology was reviewed by the researchers' committee and approved by the Ethics Committee of the Wellbeing Services County of Pirkanmaa, Finland (study ID: R230077L), the Finnish Medicines Agency, and subsequently by the institutes participating in this study. The data collection, processing, and analyses are compliant with the European General Data Protection Regulation. Data use and sharing adhere to the data governance policies of the data controller (Tampere University Hospital). The timeline of the study is illustrated in Fig 2. It is worth noting that initial analyses of images and tissue samples will take place as soon as data have been collected from the first 10 resected samples.

### Participants

Females living in the catchment area of the Wellbeing Services County of Pirkanmaa are eligible for the study if they meet all the inclusion criteria and no exclusion criteria (Table 1). Trial participants will undergo mastectomy/mastectomies based on clinical indications ranging from non-malignancy-related factors (e.g., masculinizing chest reconstruction) to risk-reducing surgery for high-risk mutation carriers and removal of cancer-affected breast(s). This study will not affect the surgical treatment predetermined for the participants as part of their standard of care treatment. Delegated surgeons will identify suitable candidates via interviews and the review of medical records. The surgeon will inform the potential participant about this study and provide a copy of the Participant Information Sheet. Informed consent will be collected prior to any protocol-specific procedure. The participant may withdraw their consent at any point.

We will recruit a maximum of 100 patients with breast cancer and 100 patients without breast cancer who are due to undergo uni- or bilateral mastectomy. Due to the exploratory nature of this investigation and the fact that insufficient prior knowledge prevents formal statistical calculations, no statistical techniques were used to determine the sample size. To allow the evaluation of the primary and secondary objectives of the study, sample size determination has been primarily based on feasibility and pragmatic considerations of the anticipated recruitment rates. We will include patients with healthy breast tissue, as we are interested in the applicability of our results to screening mammography, which encounters mainly healthy breasts.

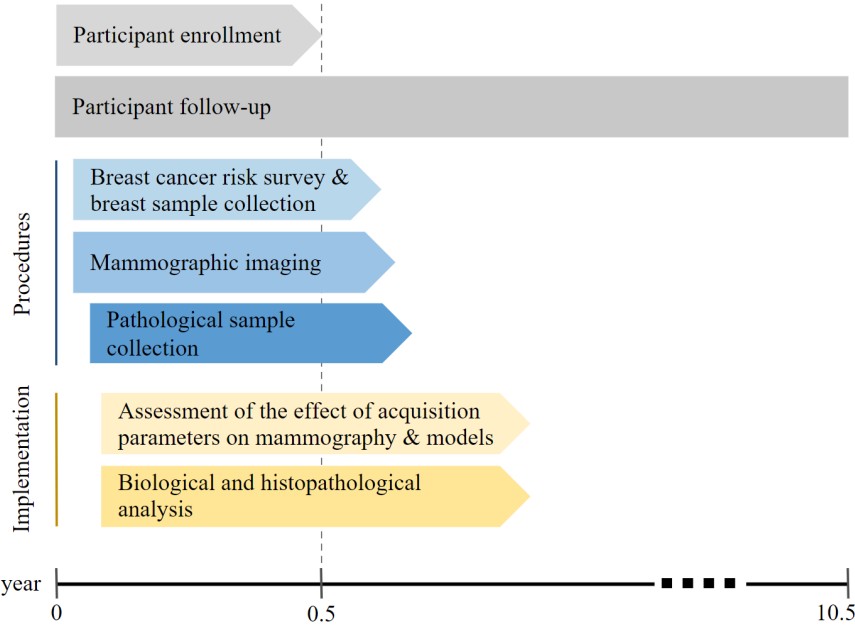

**Fig 2. Timeline of the study.** It starts with the recruitment of the first patient. Recruitment started on July 1st, 2024, and it is expected to end on December 15th, 2024. Participants will be followed up for 10 years. The imaging and sample collection from the resected samples starts with the first surgical intervention and ends soon after the last intervention. Imaging and biological analysis will begin after data have been obtained from the first 10 resected breasts and will continue after the imaging and sample collection have been completed. All analyses pertaining to this project are expected to be concluded by the 2-year mark after the imaging and sample collection have been finalized.

**Table 1. Inclusion and exclusion criteria.**

| Inclusion criteria | Exclusion criteria |
|---|---|
| Aged ≥ 18 | Aged < 18 |
| Biological female | Has breast implants |
| Ability to give informed consent | Inability to give informed consent |
| Clinical indication for uni- or bilateral mastectomy | |

## Procedures

After enrolment, participants will complete a questionnaire on breast cancer risk. Subsequently, they will undergo mastectomy according to the clinical indications. We will scan the mastectomy sample with mammography and extract radiomic features used in both classical machine learning and DL. Specimens from fibroglandular tissue, fat, and, in cases of cancer, from the tumor and axillary lymph nodes will be collected. Participants will be followed-up for 10 years after enrolment to evaluate short- and long-term mortality.

**Breast cancer risk survey.** The trial participants will complete a questionnaire on their background information regarding comorbidities and cancer risk. The researchers will collect register-based clinical data from the healthcare records. These data will be used to determine individuals' breast cancer risk scores within 5 and 10 years using the Gail model [47] and the Tyrer–Cuzick risk calculator [48], respectively.

**Mammographic imaging of the resected breast samples.** Optimal mammographic acquisition parameters are contingent on several factors: the size and density of the breast,

along with patient tolerance, determine the compression thickness and force. In addition, the x-ray tube voltage peak (kVp) can be set manually or automatically via the automated exposure control (AEC), of which there are several modes. Milliampere-seconds (mAs) are controlled by the AEC but can also be specified by manual control.

Fig 3 summarizes the study pipeline and data that will be derived from the resected tissue. The resected breast samples will undergo imaging using two mammography devices prior to further procedures. Since imaging parameters are a central variable of our study, each breast sample will be imaged with different combinations of imaging parameters, yielding several images. The acquisition parameters that will be studied are the compressed breast thickness, compression force, kVp, and mAs. We will vary these parameters based on a predefined parameter matrix structured by fixed intervals of kVp and mAs; within each position of this matrix, values of breast thickness and compression force will be used according to the capacity of each specific tissue. For each breast, we will explore a range of parameters covering the clinically used distribution and overlapping with the range of parameters used for all other samples. As there is no available literature on the mammography imaging of *ex vivo* breast tissue, we will use the clinically used compression force as a guideline, but the specific values used for breast compression force and breast thickness will be decided on a breast basis, considering the limitations of each specimen. Other system-specific hardware, such as the target-filter combination and detector, will also be considered for analysis and hence retrieved. Any further information derived from the acquisition process, such as the average glandular dose, will also be retrieved and considered for analysis.

The acquired mammograms with their corresponding tags will be stored in digital imaging and communications in medicine (DICOM) format, in FOR PROCESSING (raw) and

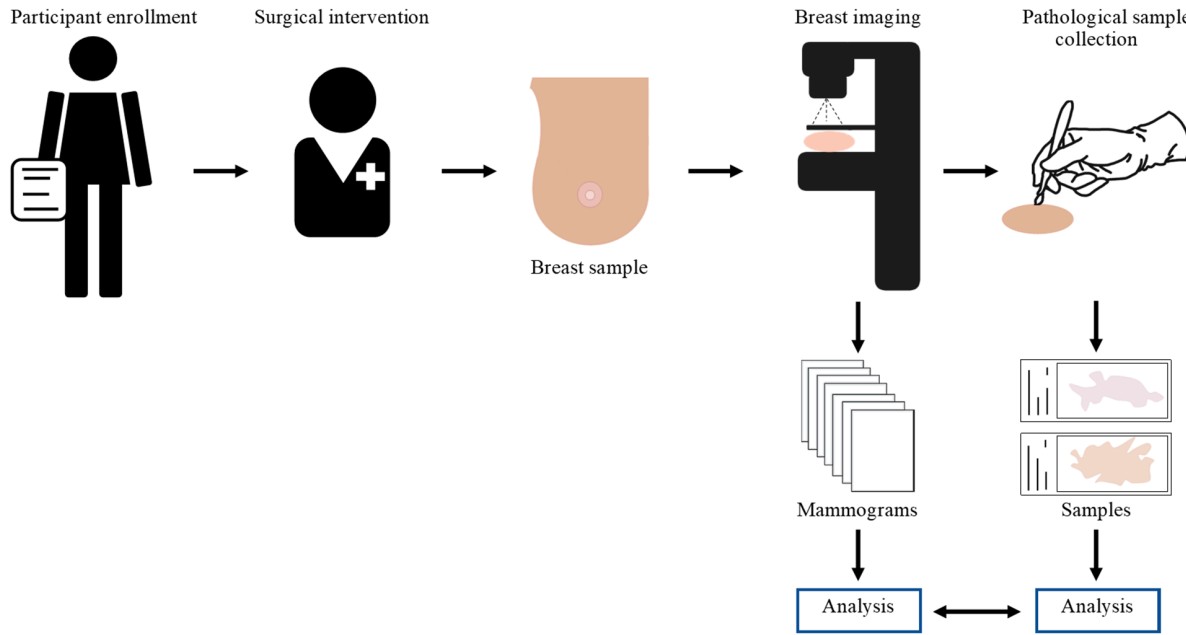

**Fig 3. Study pipeline.** Study pipeline and data that will be acquired from the resected breasts. Each resected breast will be scanned with two mammographic imaging machines with several predefined sets of imaging parameters, yielding several images per breast sample in both raw and postprocessed formats. Next, pathological samples will be extracted from the resected breasts for further biological analyses.

FOR PRESENTATION (postprocessed) formats. Mammograms will be stored in appropriate storage devices.

**Biological and histopathological samples.** After the imaging of the resected breast(s), the research team will collect representative samples of fat, fibroglandular tissue and skin, and potential tumor(s) and snap-freeze them in liquid nitrogen and/or fix them in formalin for further tissue analyses.

## Implementation

Our study will produce a multiparametric imaging dataset of mammograms. We will use this dataset to measure the effect of acquisition parameters on mammographic texture features, as well as for the validation of open-source AI models for breast cancer detection and risk assessment.

**Measuring the effect of acquisition parameters on mammographic texture features.** We will evaluate the effect of acquisition parameters on mammograms via the assessment of their texture features, by testing both the sensitivity (in terms of statistically significant differences) and the robustness of texture features in different inter- and intra-machine scenarios. Testing for statistically significant differences between scenarios allows one to determine the magnitude of the impact of the specific parameter under observation; testing for robustness is a complementary approach measuring texture feature sensitivity to imaging parameter variation. We will study radiomic texture features that are common in mammography analysis [49].

We will follow assessment approaches found in the literature on mammography acquisition effects and texture feature robustness assessment [43–46]. The metrics used for differences and robustness assessment will include the correlation coefficient, concordance correlation coefficient, paired differences, variance, means of feature ratios, percentage coefficient variation, and the Kolmogorov–Smirnov test.

**Measuring the effect of acquisition parameters at the model level.** Due to rapid technological advances, we are also interested in measuring the effect that the acquisition technique might have on the performance of classical machine learning-based and DL-based models. The first step in this phase of the project is the identification of open-source algorithms published in the field of mammography analysis regarding breast cancer detection. The identified algorithms will be subjected to further development to allow for their use and external validation with our dataset; however, the nature of our dataset allows for the establishment of different validation scenarios beyond using only the images acquired with the clinically optimal acquisition settings. The possibility of testing different "versions" of a dataset acquired from the same breast cohort allows us to study the effect that the acquisition settings can have at the model level.

Furthermore, to the extent that the identified open-source models allow, we will follow the approaches presented in two previous publications that quantified the effect of model robustness to acquisition parameters and vendor using breast phantom mammography data [46] and retrospective mammography data [45]. The first approach (pipeline A) is a two-step scheme: the authors used breast phantom imaging data and self-defined assessment metrics to assess robustness, followed by the evaluation of a breast cancer risk assessment model by using groups of features with different levels of robustness. The statistical significance of the different performances obtained was assessed via a linear mixed-effects model. The second approach (pipeline B) used retrospective screening mammograms obtained with two different mammography machines. Each subject had been previously imaged with both vendors in different screening rounds. Texture feature robustness was assessed via statistics describing

equivalence, correlation, and sample distributions, and the redundancy of features was identified via clustering. To assess the effect of robustness and redundancy on the performance of a robustness assessment model, an inter-vendor scheme was constructed by training the model using one of the vendors as a reference sample and testing it on the images of the other vendor.

Pipelines A and B will be used as guidelines to assess acquisition effects at both the texture feature and the model level. The main adjustments will be the use of *ex vivo* tissue instead of a breast phantom in the case of the first study and testing intra-acquisition settings scenarios in addition to intra-machine in the case of the second study.

**Biological and histopathological analysis.** Biological, histopathological, and genetic analyses and imaging studies will be conducted on the tissue samples to determine their structural, biological, histopathological, and genetic features to study how biological factors across different scales correlate with mammographic texture features. The tissue analyses will include, for example, routine hematoxylin-eosin staining for tissue morphology, label-free second harmonic generation imaging or Masson trichrome staining for collagen in fibroglandular tissue, cytokeratin staining for glandular epithelium, and fluorescent lipid staining for adipocytes. Tissue stainings will be analyzed using microscopic and/or mesoscopic imaging combined with modern digital pathology methods, including AI-driven image analysis.

## Ethics and dissemination

We sought a favorable opinion from the Ethics Committee of the Wellbeing Services County of Pirkanmaa (study ID: R23077L) and an organizational permission to conduct the study. The data will be pseudonymized at source. This study does not affect the clinical treatment of the participants. There will be no safety reporting on this study, since the procedures will be conducted on resected breast tissue; hence, the study does not represent risks for humans.

The study results will be published in peer-reviewed scientific journals, conference proceedings, or as part of science communication. Data use and sharing adhere to the data governance policies of the data controller (Tampere University Hospital, Finland).

## Discussion

This study aims to test whether mammography imaging acquisition introduces a batch effect and its potential impact in computerized analysis. To achieve our goals, this study will use mammography images derived from resected breast tissue, which raises questions about the imaging procedure. The relative forces experienced by the components of an *ex vivo* sample differ from those of *in vivo* tissue due to the lack of attachment to the pectoralis wall in the *ex vivo* scenario. The behavior of the resected tissue during breast compression is unpredictable, as there is no available information about this procedure. We anticipate that this will limit the number of breast compression forces and thicknesses that we will be able to use and achieve during imaging. Furthermore, the discrepancies in tissue properties that pertain to the interaction of *in vivo* and *ex vivo* breast tissue with x-rays have not been studied. At the analysis level, AI-based mammographic approaches found in the literature have been designed for images acquired from *in vivo* breasts. The extent to which the use of resected breasts will restrict the direct translation of our results to an *in vivo* scenario is impossible to predict. We will assess the differences in mammograms acquired from *ex vivo* tissue at both the quantitative level, comparing their mammographic features with those of *in vivo*, retrospective counterparts (retrieved from open-source repositories), and at the qualitative level, via radiologists' visual assessment of their likeness to an *in vivo*-acquired mammogram. Furthermore, this study will enable determination of the critical features for mammograms,

the relevance of which for *in vivo* mammography settings can further be tested in prospective patient studies.

The results of this study will shed light on one of the main factors responsible for the lack of generalizability of AI-based models. Knowing the extent and ways in which acquisition parameters affect the mammographic image at the quantitative level, and ultimately the AI models based on them, will guide future approaches in addressing the batch effect as a composite problem, instead of a homogeneous issue.

## Acknowledgments

We are grateful to the colleagues who have shared their knowledge and insight in preparation of this manuscript.

## Author contributions

**Conceptualization:** Nicole Hernández, Tomppa Pakarinen, Said Pertuz, Irina Rinta-Kiikka, Otso Arponen.

**Data curation:** Otso Arponen.

**Formal analysis:** Said Pertuz, Otso Arponen.

**Funding acquisition:** Otso Arponen.

**Investigation:** Nicole Hernández, Said Pertuz, Otso Arponen.

**Methodology:** Nicole Hernández, Tomppa Pakarinen, Annukka Salminen, Said Pertuz, Irina Rinta-Kiikka, Otso Arponen.

**Project administration:** Said Pertuz, Otso Arponen.

**Resources:** Otso Arponen.

**Software:** Otso Arponen.

**Supervision:** Said Pertuz, Irina Rinta-Kiikka, Otso Arponen.

**Validation:** Otso Arponen.

**Visualization:** Otso Arponen.

**Writing – original draft:** Nicole Hernández, Said Pertuz, Otso Arponen.

**Writing – review & editing:** Nicole Hernández, Tomppa Pakarinen, Annukka Salminen, Santiago Laguna Castro, Ulla Karhunen-Enckell, Markus Hannula, Ritva Heljasvaara, Jari Hyttinen, Katriina Joensuu, Otto Jokelainen, Arja Jukkola, Sanna-Maria Karppinen, Auni Lindgren, Eero Lääperi, Emilia Peuhu, Taina Pihlajaniemi, Renata Prunskaite-Hyyryläinen, Kirsi Rilla, Pekka Ruusuvuori, Leena Latonen, Teemu Tolonen, Masi Valkonen, Mira Valkonen, Miska Vuorlaakso, Said Pertuz, Irina Rinta-Kiikka, Otso Arponen.

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
