## [Decision Letter · Decision Letter 0]

18 Oct 2024

PONE-D-24-30561A systematic approach to study the effects of acquisition parameters and biological factors on mammographic texture features using ex vivo human tissue: a protocol descriptionPLOS ONE

Dear Dr. Hernández,

Thank you for submitting your manuscript to PLOS ONE. After careful consideration, we feel that it has merit but does not fully meet PLOS ONE’s publication criteria as it currently stands. Therefore, we invite you to submit a revised version of the manuscript that addresses the points raised during the review process.

After a careful review, we acknowledge that there are some merits in your study, particularly the relevance of the topic and the potential contribution to the field. However, the manuscript in its current form requires **major revisions** before it can be considered for publication.

We look forward to receiving your revised manuscript.

Kind regards,

Azhar Imran, Ph.D

Academic Editor

PLOS ONE

Additional Editor Comments:

Thank you for submitting your manuscript for review. After careful consideration, the reviewers have provided their feedback, and there are several major revisions required before the manuscript can be reconsidered for publication

Reviewers' comments:

Reviewer's Responses to Questions

**Comments to the Author**

1. Does the manuscript provide a valid rationale for the proposed study, with clearly identified and justified research questions?

Reviewer #1: Yes

Reviewer #2: Partly

2. Is the protocol technically sound and planned in a manner that will lead to a meaningful outcome and allow testing the stated hypotheses?

Reviewer #1: Yes

Reviewer #2: Partly

3. Is the methodology feasible and described in sufficient detail to allow the work to be replicable?

Reviewer #1: Yes

Reviewer #2: No

4. Have the authors described where all data underlying the findings will be made available when the study is complete?

Reviewer #1: Yes

Reviewer #2: Yes

5. Is the manuscript presented in an intelligible fashion and written in standard English?

Reviewer #1: Yes

Reviewer #2: Yes

6. Review Comments to the Author

You may also provide optional suggestions and comments to authors that they might find helpful in planning their study.

Reviewer #1: The authors proposed a protocol to collect a new dataset for investigating the effects of biological and non-biological heterogeneity on the mammograms. The research idea is original and reasonable. In general, the manuscript is carefully written. The aims of the protocol were explained in detail. Here are my suggestions:

1) Figure 1 is a little bit misleading. The biological factors should be on top and the arrows should be directed from the factors to the mammographic texture features. In addition, imaging parameters should be given as a list separately. An arrow should be directed from the list to the mammographic texture features. This may be more informative for the readers.

2) The authors should explain why they used “dense” in line 171.

3) In Figure 3, the “man symbol” may be replaced by a “woman symbol” since most of the participants will be women.

Reviewer #2: This is an important study on a very interesting subject matter. I think however that the background and aims are hard to follow, and the study as a whole could benefit from a more focused narrative and more clearly explained purpose.

ABSTRACT

The background section in the abstract should be more concise and focus on what you are actually going to explore in the study.

BACKGROUND

Line 21-22: Calling breast cancer one of the most lethal cancers world wide is true in an absolute population sense, but relatively speaking and for the individual it is certainly not among the most lethal with ~95% five-year survival. Please rephrase.

Line 22-29: It would be helpful for readers that are less familiar with breast cancer and mammography if you could provide some numbers backing up your points here, e.g. how much does screening reduced mortality, what is the percentage of breasts that are considered dense, how much does the risk depend on density and what is the sensitivity in various density groups.

Line 32-39: This definition of radiomics is not one I am familiar with. Radiomics is normally either a set of specific standardized quantitative features of radiological images that have been found to correlate with various genomic factors, or sometimes any quantitative image features when used in this manner. Radiomics has no direct relationship to machine learning, and indeed one could call the entire idea of using AI-analysis of images to find biomarkers as simply AI-driven radiomics.

Line 41-42 While this statement could be true (from a certain point of view) two references are from 2020 and 2018 and the third is a self-citation. What is “often”? There are at least two examples of commercial cancer detection software for mammography that has been shown through randomized clinical trials to decrease workload and increase cancer detection when used in breast cancer screening. There are also several examples of other similar software performing well on external independent validation set.

Lines 42-45 Are the statements about high heterogeneity meant to be backed up by refs 26-28.

Lines 49-52 I don’t understand this reasoning. The point of radiomics is to be able to get biomarkers from images and not from biological samples. This seems like circular reasoning to me. We are imaging the biology of the breast, so of course the biology will affect the radiology, how else would anything be detectable? The biological properties of the breast do not affect the textural features, they ARE the textural features (in combination with the radiological imaging technique of course).

Lines 52-54 What research is scarce? Into the breast micro-tumour environment?

Lines 56-59 What would be the specific purpose of having non-optimal acquisition performance for validation? Also, degrading image quality to accurately simulate dose reduction is relatively straight-forward to do retrospectively.

Lines 69-72 I think these goals are too non-specific. I understand that it is an exploratory study, but it would benefit from some kind of hypothesis and defined quantitative (or even qualitative) endpoints and figures of merit.

METHODS

General comment: I have two general concerns about the Methods and design.

1. What is the role of AI in the project? It is neither in the title nor in the objectives, but is mentioned throughout the paper and implied to be important. However, there is no mention in the methods of what kind of AI software or what kind features are going to be tested. You seem to combine AI for cancer detection and for risk assessment, two tasks which are quite different from each other.

2. How are you going to vary acquisition parameters? Is there a set protocol for this with certain increments of tube voltage and mAs? Will this be different for different vendors? How will you compensate for the differences from real clinical images and the mastectomy samples that you will employ? Especially considering that the lack of chest wall attachment will result in a very different level of compression and thickness which will in turn likely make it unsuitable to use the automatic exposure control. Will all breasts be imaged across the same range of setting to provide comparable images?

Lines 151-153 What does it mean that you will “ reach out to include as much overlap of parameter settings between individual breast as possible”?

Lines 209-212 While biological-radiological correlations are hailed as the main feature of the study, there is very little detail on the biological and histopathological analysis. What exactly are you going to analyse in the benign adipose and fibroglandular tissue samples? I don’t see how you could e.g. analyse the structural composition of the breast from small tissue samples.

7. PLOS authors have the option to publish the peer review history of their article (what does this mean?). If published, this will include your full peer review and any attached files.

Reviewer #1: No

Reviewer #2: No

---

## [Author Response · Author response to Decision Letter 1]

5 Dec 2024

Dear editor and reviewers,

Thank you for your insightful comments on our manuscript. We have addressed your suggestions and we believe they have improved the clarity, detail and focus of our work. We have submitted a point-by-point response letter, outlining the changes in detail, please refer to this for specific responses.

---

## [Decision Letter · Decision Letter 1]

27 Mar 2025

PONE-D-24-30561R1

A systematic approach to study the effects of acquisition parameters and biological factors on computerized mammography analysis using ex vivo human tissue: A protocol description

PLOS ONE

Dear Dr. Hernández,

I am writing with an update about your submission, "A systematic approach to study the effects of acquisition parameters and biological factors on computerized mammography analysis using ex vivo human tissue: A protocol description" (PONE-D-24-30561R1).  During the submission process] you requested several  authorship changes. The nature, extent and timing of the requests to change the author list call into question whether your manuscript complies with the PLOS Authorship policy (https://journals.plos.org/plosone/s/authorship). As such, we are rescinding the Accept decision and rejecting the manuscript.  We may reconsider this submission in the future if a research integrity official at the corresponding author’s institution reviews and provides verification of the article's authors and their contributions. The following documents would need to be provided when resubmitting:a.      Written, signed statements from all contributors, including added/removed authors, confirming that all agree with the article’s author list and contributionsb.      Cover letter that describes the contribution of each author and provides a specific reason why each author was added or removed after initial submissionc.      Formal letter from a research integrity official or equivalent at the corresponding author’s institution, or the institution where the majority of the research was conducted, confirming the author list and stated contributions.d.      Institutional email address for the official responsible for oversight of research and/or research integrity at the corresponding author’s institution.  I am sorry we do not have more positive news, but hope that you understand the reasons why we rejected this submission.

Kind regards,

Joanna Tindall, PhD

Staff Editor

PLOS ONE

Additional Editor Comments (if provided):

Reviewers' comments:

Reviewer's Responses to Questions

**Comments to the Author**

1. Does the manuscript provide a valid rationale for the proposed study, with clearly identified and justified research questions?

Reviewer #1: Yes

Reviewer #2: Yes

2. Is the protocol technically sound and planned in a manner that will lead to a meaningful outcome and allow testing the stated hypotheses?

Reviewer #1: Yes

Reviewer #2: Yes

3. Is the methodology feasible and described in sufficient detail to allow the work to be replicable?

Reviewer #1: Yes

Reviewer #2: Yes

4. Have the authors described where all data underlying the findings will be made available when the study is complete?

Reviewer #1: Yes

Reviewer #2: Yes

5. Is the manuscript presented in an intelligible fashion and written in standard English?

Reviewer #1: Yes

Reviewer #2: Yes

6. Review Comments to the Author

You may also provide optional suggestions and comments to authors that they might find helpful in planning their study.

Reviewer #1: I congratulate the authors since they have addressed all of the remaining issues. My suggestion is that the authors report the thickness of the tissues during imaging.

Reviewer #2: Thank you for your careful edits, I think that they clarify and answer all the issues I raised in my earlier review.

7. PLOS authors have the option to publish the peer review history of their article (what does this mean?). If published, this will include your full peer review and any attached files.

Reviewer #1: No

Reviewer #2: **Yes: **Magnus Dustler

- - - - -

---

## [Author Response · Author response to Decision Letter 2]

4 May 2025

We addressed the reviewer and editor's comments point by point in the requested response letter.

---

## [Editor Report · Decision Letter 2]

17 Jul 2025

A systematic approach to study the effects of acquisition parameters and biological factors on computerized mammography analysis using ex vivo human tissue: A protocol description

PONE-D-24-30561R2

Dear Dr. Hernández,

We’re pleased to inform you that your manuscript has been judged scientifically suitable for publication and will be formally accepted for publication once it meets all outstanding technical requirements.

Kind regards,

Yuchen Qiu, Ph.D.

Academic Editor

PLOS ONE
---

## [Editor Report · Acceptance letter]

PONE-D-24-30561R2

PLOS ONE

Dear Dr. Hernández,

I'm pleased to inform you that your manuscript has been deemed suitable for publication in PLOS ONE. Congratulations! Your manuscript is now being handed over to our production team.

Kind regards,

on behalf of

Dr. Yuchen Qiu

Academic Editor

PLOS ONE